# Role of Smartphone Applications in the Assessment and Management of Fatigue in Patients with Multiple Sclerosis: A Scoping Review

Annibale Antonioni [1,2] , Andrea Baroni [1,3,*] , Giada Milani [4] , Irene Cordioli [1] and Sofia Straudi [1,3]

1 Department of Neuroscience and Rehabilitation, University of Ferrara, 44121 Ferrara, Italy; annibale.antonioni@edu.unife.it (A.A.); irene.cordioli@edu.unife.it (I.C.); strsfo@unife.it (S.S.)
2 Doctoral Program in Translational Neurosciences and Neurotechnologies, University of Ferrara, 44121 Ferrara, Italy
3 Department of Neuroscience, Ferrara University Hospital, 44124 Ferrara, Italy
4 Center for Translational Neurophysiology of Speech and Communication (CTNSC), Italian Institute of Technology (IIT), 44121 Ferrara, Italy; mlngdi@unife.it
* Correspondence: brnndr3@unife.it; Tel.: +39-0532-236185

**Abstract:** Fatigue is a common symptom in Multiple Sclerosis (MS), and its assessment depends entirely on patient reports. Importantly, managing MS symptoms is increasingly supported by Digital Health Technology (DHT), which includes Mobile Health Technology (mHT). Considering the growing interest, we aimed to synthesise evidence about smartphone applications for the assessment and management of fatigue in MS, as well as to investigate their usability, feasibility, and reliability. We performed a literature search in PubMed, Science Direct, and Embase using a scoping review approach. We included 16 articles and, although many lacked crucial methodological details, DHT was evaluated in all MS clinical subtypes and with disease durations up to more than 20 years. Despite the marked heterogeneity in terms of the employed methods, all documented a high degree of usability, assessed both as feedback from participants and completed tasks. Moreover, the feasibility assessment also showed good results, as apps were able to discriminate between patients with and without fatigue. Importantly, most also showed excellent results in terms of reliability, and some patients reported a reduction in fatigue thanks to mHT. Despite limitations, mHT has been positively evaluated by patients, suggesting a promising role of DHT in the self-management of MS.

**Keywords:** fatigue; digital health technology (DHT); mobile health technology (mHT); multiple sclerosis (MS); home self-management; usability; feasibility; reliability; smartphone applications

## 1. Introduction

Fatigue is one of the most common symptoms in Multiple Sclerosis (MS), reported by at least 75% of patients and considered one of the main causes of impaired quality of life (QoL) [1]. Even if a motor and cognitive domain has been identified in the physiopathology of fatigue [2], no objective measures are available to clearly identify perceived fatigue, which depends entirely on the patient's perception and report [3]. Currently, several fatigue scales are available for measuring the severity, intensity, and characteristics of experienced fatigue and/or fatigue impact on QoL, each with advantages and limitations [1,4].

Currently, treatment for MS includes disease-modifying therapies and symptom management strategies [5], and both pharmacological and non-pharmacological interventions are recommended [6]. Managing MS symptoms is increasingly supported by Digital Health Technology (DHT) [5]. Indeed, DHTs can improve patient engagement in self-care, encourage positive physical functioning, support psychological self-management, reduce communication gaps, record and share health-related information, and personalize services to meet patient needs [7,8].

The DHT includes Mobile Health Technology (mHT), defined as "the use of mobile and wireless devices to improve health outcomes, healthcare services, and health research" [9], particularly through the use of smartphone applications ("Apps") [10]. Although there are many types of mHTs in this field, they are generally commercially available smartphone applications that allow patients to actively participate in the management of their disease, not only by monitoring their daily routine through reports and diaries but also by receiving treatment reminders and specific lifestyle advice from specialists [11]. In addition, they often provide access to simple exercises and games designed not only to assess one's performance over time but also to improve it in a simple, fun, and engaging way for the patient [12]. However, the integration of mHT in MS management is not always obvious. The usability, i.e., the degree of patient satisfaction and ease of use, of every device needs to be shown, in order to demonstrate how they could be user-friendly or complex for the patient [7]. Importantly, mHT appeared to be a dominant concept in the field of DTH [13], and many PwMS regularly have access to mobile technology and use devices like a smartphone or tablet [9]. Coherently, many smartphone applications for PwMS are commonly used, and this field has undergone a steady growth in recent years, although, at least at the moment, the dissemination of these technologies is mainly limited to Western countries, i.e., those with the economic resources and educational levels to support their implementation [13,14]. However, given the extreme heterogeneity in terms of recruited patients, study designs, and mHTs used, conclusive evidence on their applicability in PwMS is still lacking.

Considering the growing research interest, this scoping review aims to report evidence about the use of smartphone applications in the assessment and management of fatigue in PwMS. We also investigate the usability, the feasibility, i.e., the applicability and generalisability of the home use of the apps, and the reliability, i.e., the ability of the apps to reduce fatigue or improve home self-management through exercises and/or strategies to stimulate and guide the patient in daily life, of the proposed new technologies.

## 2. Materials and Methods

The protocol of this scoping review was pre-registered on Open Science Framework (OSF) with DOI https://doi.org/10.17605/OSF.IO/C4V2Y. We applied the PRISMA-ScR framework [15]. The aims of this scoping review are as follows:

- Search, identify, and synthesise research on mobile applications for the assessment and management of fatigue in PwMS.
- Categorize the areas of assessment and/or intervention in psychological/cognitive and motor/functions.
- Describe ability of mobile applications to reduce fatigue.
- Describe users' satisfaction and usability.
- Provide clinicians with a practical overview of the use of mHT in the assessment and management of fatigue in PwMS, highlighting the mobile applications that currently seem to be the most promising for this purpose.

### 2.1. Search Strategy

Articles published in peer-reviewed journals, published conference proceedings, and pre-peer review web publications were potentially eligible. Authors GM and IC conducted literature searches of electronic bibliographic databases in Web of Science, PubMed, Science Direct, and Embase from 2012 to 2022 inclusive. The database search was completed on 30 June 2021 and frequently updated until the date of submission. The research strategy incorporates controlled vocabulary and keywords (i.e., Multiple Sclerosis, mobile application, fatigue); for a complete research strategy, see Supplementary File S1. We allowed for any smartphone applications that evaluate, monitor, and treat the fatigue symptoms in PwMS. For the MS reference standard, any diagnostic process was accepted (i.e., MRI/A, and/or specialist opinion).

*2.2. Study Inclusion/Exclusion Criteria*

We included all studies on PwMS (with no limits on age and phenotype) suffering from fatigue and who have used smartphone applications to manage their symptoms.

We included observational (case–control studies and cohort studies) and interventional studies that applied smartphone application data as potential instruments to assess and treat fatigue.

Inclusion criteria were as follows: (i) reference in English; (ii) reference includes original quantitative and qualitative data; (iii) study includes PwMS. Exclusion criteria were as follows: (i) studies of patients who do not consider fatigue as an outcome; (ii) commentaries, editorials, and other publication forms without primary data (scoping reviews, systematic reviews).

*2.3. Study Selection*

Data were independently extracted by two reviewers (GM and IC) and discrepancies solved via group discussion. A data extraction framework was developed and included the following: author(s), year of publication, country of origin, study design, population, type of MS, EDSS, average duration of MS, application name, online stores, app features (time of exploring language, description, function), degree of users' satisfaction, and usability evaluation method. According to each study design, the results related to usability, feasibility, and reliability were reported. For usability, the degree of patient's satisfaction and ease of app use were evaluated. For feasibility, we recorded the applicability and generalisability of domestic use of the apps. Finally, for the reliability evaluation, we analysed the capability of the apps to reduce fatigue or improve its management.

As this was a scoping review, there was no a priori plan for data meta-analysis, and a narrative description is provided. Data are presented in tables according to reference standard or outcome measure following publication date.

**3. Results**

Database searches identified 205 records. After duplicate removal, 161 records were screened for titles and abstracts, and 123 records were excluded. The remaining 38 full-text articles were assessed, and 22 of them were excluded: 3 studies did not meet the study design, and 19 studies evaluated a different outcome. After full-text evaluation, 16 articles were included for data extraction. All of them were observational studies.

All the studies were published in the last ten years, with a wide geographic distribution: seven in the USA [16–22]; three in Belgium [23–25] and the UK [25–27]; two in New Zealand [26,27] and Switzerland [17,28]. The remaining studies were from Hungary, Spain, Finland, Iran, and Netherlands [18,29–31]. Using available data (one study did not report the characteristics of the sample [31]), a total number of 987 PwMS were recruited. The mean number of PwMS involved in each study was 67 (range 10–495); of them, 40 (range 6–232) were female. The mean age of participants ranged between 35.1 and 60.97 years. The mean Expanded Disability Status Scale (EDSS) score varied from 2.1 to 5.3. The average duration of disease varied from 3 to 21.48 years, but seven studies did not report information about illness duration [17,20,23,24,28,29,31]. Primary Progressive (PPMS), Relapsing Remitting (RRMS), Secondary Progressive (SPMS), and Clinical Isolated Syndrome (CIS) types of MS were included in the studies. The most frequent type of MS was RRMS; seven studies did not provide information on the type of MS included [17,24,26–29,31]. In Table 1, the study reference (i.e., title, authors, year of publication, study design, country, journal) and the characteristics of the sample (i.e., sample size, type of MS, average duration of illness, EDSS) are reported.

**Table 1.** Study reference and characteristics of the sample.

| Unique Identifying Number | Title | Author | Year of Publication | Study Design | Country | Journal | Sample Size | Type of MS | Avarage Duration of Illness | EDSS |
|---|---|---|---|---|---|---|---|---|---|---|
| 1 | Design considerations for a multiple sclerosis fatigue mobile app MS Energize: A pragmatic iterative approach using usability testing and resonance checks | van Kessel, K; et al. [27] | 2021 | Observational | United Kingdom (UK), New Zealand (NZ) | INTERNET INTERVENTIONS-THE APPLICATION OF INFORMATION TECHNOLOGY IN MENTAL AND BEHAVIOURAL HEALTH | 11 patients (7 females; mean age, 47.25; range, 40–54 | N/A | 11 years | N/A |
| 2 | Usability of a Mobile App for Real-Time Assessment of Fatigue and Related Symptoms in Patients With Multiple Sclerosis: Observational Study | Palotai, M; et al. [18] | 2021 | Observational | United States (USA), Hungary (HU) | JMIR MHEALTH AND UHEALTH | 64 patients (54 females; mean age, 52) | 58 RRMS; 5 SPMS; 1 CIS | 20 years | 2.1 |
| 3 | More Stamina, a Gamified mHealth Solution for Persons with Multiple Sclerosis: Research Through Design | Giunti, G; et al. [31] | 2018 | Research Through Design | Spain, Finland | JMIR MHEALTH AND UHEALTH | N/A | N/A | N/A | <4.5 |
| 4 | A Novel Digital Care Management Platform to Monitor Clinical and Subclinical Disease Activity in Multiple Sclerosis | Van Hecke, W; et al. [25] | 2021 | Observational | Belgium, United Kingdom | BRAIN SCIENCES | feasibility study: 45 patients (36 females; mean age, 45.6) | 25 RRMS; 7 SPMS; 5 PPMS; 8 unknown | Range, 3–20 years | N/A |

**Table 1.** *Cont.*

| Unique Identifying Number | Title | Author | Year of Publication | Study Design | Country | Journal | Sample Size | Type of MS | Avarage Duration of Illness | EDSS |
|---|---|---|---|---|---|---|---|---|---|---|
| 5 | Evaluating the Utility of Smartphone-Based Sensor Assessments in Persons With Multiple Sclerosis in the Real-World Using an App (elevateMS): Observational, Prospective Pilot Digital Health Study | Pratap, A; et al. [22] | 2020 | Observational (Prospective and cross-sectional) | United States (USA) | JMIR MHEALTH AND UHEALTH | 629 subjects (259 females). 495 pwMS (232 females, mean age, self-referred 45.20—clinic-referred 48.93); 134 healthy controls (mean age, 39.34) | 423 RRMS; 30 SPMS; 40 PPMS; 2 unknown | Range, 11.14–14.29 years | N/A |
| 6 | Testing Feasibility of a Mobile Application to Monitor Fatigue in People With Multiple Sclerosis | Newland, P; et al. [16] | 2019 | Observational | United States (USA) | JOURNAL OF NEUROSCIENCE NURSING | 32 patients (26 females; mean age, 49) | 30 RRMS; 2 SPMS | 11.2 years | 3 |
| 7 | Smartphone-Based Tapping Frequency as a Surrogate for Perceived Fatigue. An in-the-Wild Feasibility Study in Multiple Sclerosis Patients | Barrios, L; et.al. [17] | 2021 | Observational | Switzerland, United States (USA) | MULTIPLE SCLEROSIS JOURNAL | 35 patients (20 females; mean age, 36.77; range, 21–53) | N/A | N/A | 2.31 (range, 0–6) |

**Table 1.** *Cont.*

| Unique Identifying Number | Title | Author | Year of Publication | Study Design | Country | Journal | Sample Size | Type of MS | Avarage Duration of Illness | EDSS |
|---|---|---|---|---|---|---|---|---|---|---|
| 8 | Electronic visual analogue scales for pain, fatigue, anxiety and quality of life in people with multiple sclerosis using smartphone and tablet: a reliability and feasibility study | Kos, D; et al. [24] | 2017 | Observational | Belgium | CLINICAL REHABILITATION | 52 patients (34 females; mean age, 49.1); 52 healthy controls (36 females; mean age, 47.5); range, 20–70 | N/A | N/A | N/A |
| 9 | MS Energize: Field trial of an app for self-management of fatigue for people with multiple sclerosis | Babbage, DR; et al. [26] | 2019 | Observational | New Zealand (NZ), United Kingdom (UK) | INTERNET INTERVENTIONS-THE APPLICATION OF INFORMATION TECHNOLOGY IN MENTAL AND BEHAVIOURAL HEALTH | 11 patients (6 females; mean age, 49; range, 41–59) | N/A | 9 years | N/A |
| 10 | A Smartphone-based Application for Self-Management in Multiple Sclerosis | Mokhberdezfuli, M; et al. [29] | 2021 | Observational | Iran | JOURNAL OF HEALTHCARE ENGINEERING | 60 patients (49 females; mean age, 36.8; range, 31–40); 6 healthy subjects | N/A | N/A | N/A |
| 11 | Identifying and Quantifying Neurological Disability via Smartphone | Boukhvalova, AK; et al. [21] | 2018 | Observational (prospective and cross-sectional) | United States (USA) | FRONTIERS IN NEUROLOGY | prospectives participants: 16 patients (10 females; mean age, 60.97); 15 healthy subjects | 1 RRMS; 5 SPMS; 10 PPMS | 21.48 | 5.31 |

**Table 1.** *Cont.*

| Unique Identifying Number | Title | Author | Year of Publication | Study Design | Country | Journal | Sample Size | Type of MS | Avarage Duration of Illness | EDSS |
|---|---|---|---|---|---|---|---|---|---|---|
| 12 | WalkWithMe: Personalized Goal Setting and Coaching for Walking in People with Multiple Sclerosis | Geurts, E; et al. [23] | 2019 | Observational, Case series | Belgium | ACM UMAP '19: PROCEEDINGS OF THE 27TH ACM CONFERENCE ON USER MODELING, ADAPTATION AND PERSONAL-IZATION | 13 patients (all females; mean age, 45) | All RRMS | N/A | N/A |
| 13 | Real-world keystroke dynamics are a potentially valid biomarker for clinical disability in multiple sclerosis | Lam, KH; et al. [30] | 2021 | Observational | Netherlands | MULTIPLE SCLEROSIS JOURNAL | 85 patients (64 females; mean age, 46.4); 18 healthy subjects | 51 RRMS; 25 SPMS; 9 PPMS | 11.3 years | 3.5 (range, 1.5–7) |
| 14 | Evaluating more naturalistic outcome measures A 1-year smartphone study in multiple sclerosis | Bove, R; et al. [19] | 2015 | Observational | United States (USA) | NEUROLOGY-NEUROIMMUNO LOGY & NEUROINFLAM-MATION | 38 patients (28 females; mean age, 35.1); 38 healthy subjects | 25 RRMS; 5 SPMS; 1 PPMS; 2 CIS; 5 unknown | 7.8 years | 2.3 |
| 15 | A Rapid Tapping Task on Commodity Smartphones to Assess Motor Fatigability | Barrios, L; et al. [28] | 2020 | Observational | Switzerland | PROCEEDINGS OF THE 2020 CHI CONFERENCE ON HUMAN FACTORS IN COMPUTING SYSTEMS (CHI'20) | 20 patients (11 females; mean age, 43.1; range, 20–62); 35 healthy subjects | N/A | N/A | 3 (range, 0–8) |

**Table 1.** *Cont.*

| Unique Identifying Number | Title | Author | Year of Publication | Study Design | Country | Journal | Sample Size | Type of MS | Avarage Duration of Illness | EDSS |
|---|---|---|---|---|---|---|---|---|---|---|
| 16 | Development and Validation of the FSIQ-RMS: A New Patient-Reported Questionnaire to Assess Symptoms and Impacts of Fatigue in Relapsing Multiple Sclerosis | Hudgens, S; et al. [20] | 2019 | Observational | United States (USA) | VALUE IN HEALTH | 10 patients (7 females; mean age, 42.1; range, 27–54) | 8 RRMS; 2 SPMS | N/A | 3.2 (range, 0–6) |

Abbreviations: EDSS: Expanded Disability Status Scale; RRMS: Relapsing-Remitting Multiple Sclerosis; SPMS: Secondary Progressive Multiple Sclerosis; PPMS: Primary Progressive Multiple Sclerosis; CIS: Clinical Isolated Syndrome; N/A: Not Available.

### 3.1. Classification of Mobile Applications

Eleven smartphone applications were included with their trade names: MS Energize, More Stamina, iCompanion, ElevateMS FatigueApp, Electronic VAS, MS Energize, Tapping Test and Balloon Popping Test App, WalkWithMe, Neurokeys keyboard APP, and FSIQ-RMS e-Diary. Five studies did not report the name of the application [17–19,28,29]. Most of the applications were created only for Android (*n* = 8) [17–21,23,24,29] or Apple devices (*n* = 3) [22,26,27]; three applications can be downloaded from both the Android and iOS e-store [25,28,30]; two studies did not report the e-store available for the application download [16,31]. Only six studies specified in which language the application was available: the most used was English [16,20,25–27,31], followed by Italian, Dutch, German, French, Spanish [25], and other unspecified languages [20].

Ten of the included applications allowed us to evaluate and monitor the level of fatigue over time [16–22,24,28,30]; six allowed both the evaluation and management of fatigue and its related symptoms [23,25–27,29,31]. In Table 2, the characteristics of smartphone applications are summarized.

### 3.2. Degree of Patient Satisfaction and Usability Evaluation Methods

The most used method to assess application usability was questionnaires (*n* = 6). Among these, the System Usability Scale (SUS questionnaire), the Questionnaire for User Interaction Satisfaction (QUIS questionnaire), and the user-version of the Mobile Application Rating Scale (uMARS questionnaire) were used [20,26,29]; one study used a self-made questionnaire [23]. Four studies used interviews and surveys for evaluating application usability [20,23,25,26], three studies used the task completion [16,18,22], one study used think-aloud [27], and one the heuristic evaluation [31]. A combined evaluation was used in four studies: questionnaires and interviews [20,23,26] or questionnaire and think-aloud [27]. Five studies did not report any usability evaluation [17,19,21,28,30]. In Table 3, the usability assessment is reported.

**Table 2.** Characteristic of the smartphone applications.

| Unique Identifying Number | Application Name | Online Stores | Time | Language | App Decription | Assessment | Management |
|---|---|---|---|---|---|---|---|
| 1 | MS Energize | iOS | N/A | English | App content included general information about MS fatigue, factors that may influence MS fatigue and a section on planning for the future. Each of the seven main modules contained 2–4 subsections, within which were levels of topic relevant education ('Learn'), an interactive task to engage with ('Interact') and an opportunity to apply what was learned by developing an action plan ('Apply'). The app also provided visual summaries for users and encouragement on their progress and achievements. | Yes | Yes |
| 2 | N/A | Android | 2 weeks | N/A | The mobile application features three modules: (1) A series of one-time questionnaires, (2) Visual Analogue Scales (VAS) for self-reporting of fatigue, depression, anxiety, and pain levels, (3) A sleep diary (SLD) containing a series of questions regarding perceived duration and quality of sleep, as well as the same VAS described above. Reminder functions were implemented into the mobile application. | Yes | No |
| 3 | More Stamina | N/A | N/A | English | More Stamina acts as a to-do list where users can input the tasks they want to accomplish that day in a simple manner. A person's overall energy is represented through a progress bar composed of Stamina Credits. Users start their day with 100 points or Stamina Credits and assign a certain amount of them to activities for that day. More Stamina also has a user profile feature that collects and aggregates information about the user's condition. Surveys, questionnaires, and other assessment tools such as the FSS and CFS are optionally available for completion. | Yes | Yes |

**Table 2.** *Cont.*

| Unique Identifying Number | Application Name | Online Stores | Time | Language | App Decription | Assessment | Management |
|---|---|---|---|---|---|---|---|
| 4 | iCompanion | Android; iOS | N/A | Italian, Dutch, English, German, French and Spanish | Using the iCompanion app, PwMS can keep a diary, log symptoms, and perform test for body functions, cognitive functions and fatigue based on PROs. In addition, PwMS can add treatment information and set reminders on when to take or perform their treatment, as well as learn about topics related to MS. | Yes | Yes |
| 5 | ElevateMS | Apple App Store | 12 weeks | N/A | ElevateMS primarily targeted collection of real-world data from participants with MS. This included self-reported measures of symptoms and health via optional "check-in" surveys, and independent assessments of motor function via sensor-based active functional tests. Local weather data were collected every time an assessment was performed. | Yes | No |
| 6 | FatigueApp.com | N/A | N/A | English | The FatigueApp collect data to correlate fatigue measures with other symptoms and quality of life. The 12-item World Health Organization Disability Assessment Schedule 2.011 is used to report quality of life. The 10-cm visual analog scale is used to assess pain severity. The PROMIS Fatigue Scale Short Form was used to measure MS-related fatigue severity and the PROMIS Cognitive Abilities and Cognitive Concerns scale to assess perceived cognitive dysfunction. | Yes | No |

**Table 2.** *Cont.*

| Unique Identifying Number | Application Name | Online Stores | Time | Language | App Decription | Assessment | Management |
|---|---|---|---|---|---|---|---|
| 7 | N/A | Android | 2-week | N/A | The app proposed a rapid tapping task on a smartphone as an inexpensive approach to assessing motor fatigability. Participants performed the tapping task with their dominant hand during each day. The app also sent daily notifications to the patients to remind them to complete the tapping tasks as well as to complete the FSS questionnaire once per week directly in the app. | Yes | No |
| 8 | Electronic VAS | Android | N/A | N/A | Participants completed electronic visual analogue scales on a smartphone, where fifteen statements covering the domains of fatigue, anxiety, pain and quality of life. | Yes | No |
| 9 | MS Energize | iOS iPhone | 5–6 weeks | English | The app covers MS fatigue, how to use energy effectively, how behavior, thoughts and emotions interact and impact on MS fatigue, as well as the potential effects of bodily and environmental factors. MS Energize provides education, interactive tasks, and supports application of the principles into a user's day-to-day life. | Yes | Yes |
| 10 | N/A | Android | N/A | N/A | The app includes a Patient's Panel with educational content related to MS, patient medical record, patient health status, contacting a physician, and MS care centers. Patients could enter his/her personal and clinical data to create a medical record. A patient could also send messages to his/her physician and receive a response. The app also includes a Physician's Panel. | Yes | Yes |

**Table 2.** *Cont.*

| Unique Identifying Number | Application Name | Online Stores | Time | Language | App Decription | Assessment | Management |
|---|---|---|---|---|---|---|---|
| 11 | Tapping Test; Balloon Popping Test | Android | 9 weeks | N/A | In the tapping test, users had to tap as quickly as possible over a 10 s duration and the final score is the average of two attempts. The balloon popping test expands neurological functions necessary for test completion from pure motoric, to motoric, visual, and cognitive. The primary goal for this test is to touch as many randomly generated dark blue circles (balloons) moving across the screen in succession over the 26-s test duration as possible. | Yes | No |
| 12 | WalkWithMe | Android | 10 weeks | N/A | WalkWithMe is an application that supports pwMS to achieve a self-set goal for walking over a period of 10 weeks. The application provides also audio feedback and a virtual coach, to assist the user in performing walking activities towards the target goal. Users can also log certain factors, such as fatigue, weather conditions, and surface. | Yes | Yes |
| 13 | Neurokeys keyboard App | Android; iOS | 2 weeks | N/A | Neurokeys measures health status through regular typing on the smartphone. During regular typing, keyboard interactions of interest were logged and timestamped in the background and general typing characteristics were obtained. | Yes | No |
| 14 | N/A | Android | 1 year | N/A | The application consists on a suite of 19 tests, designed to assess participant performance (color vision, attention, dexterity, and cognition), and elicit patient-reported outcomes (PROs; fatigue, mood, and QoL). | Yes | No |

**Table 2.** *Cont.*

| Unique Identifying Number | Application Name | Online Stores | Time | Language | App Decription | Assessment | Management |
|---|---|---|---|---|---|---|---|
| 15 | N/A | Any | N/A | N/A | Participants performed a rapid alternating tap on the smartphone screen to assess motor fatigability. Participants were asked to perform the tapping task as fast as possible without stopping until the app indicated completion. | Yes | No |
| 16 | FSIQ-RMS (e-Diary) | Android | N/A | English and other 45 translations | The FSIQ-RMS is a new PRO instrument to assess fatigue symptoms relevant to patients within the spectrum of RMS and the relevant impact of these symptoms on patients' lives. | Yes | No |

Abbreviations: FSS: Fatigue Severity Scale; PROs: Patient Reported Outcomes; QoL: Quality of Life; FSIQ-RMS: Fatigue Symptoms and Impact Questionnaire—Relapsing Multiple Sclerosis; N/A: Not Available.

**Table 3.** Degree of patient satisfaction and usability evaluation methods.

| Unique Identifying Number | Degree of Satisfaction | Usability Evaluation Method |
|---|---|---|
| 1 | The usability testing and resonance checks suggested that user experience of MS Energise was mostly positive, and participants provided a number of suggestions for improvements to aspects of content and design. | Think-aloud; Questionnaire SUS |
| 2 | It is reported good patient compliance with the mobile application based assessments. Patients reported no issues with the usage of the application. | Task completion |
| 3 | Several major and minor usability problems were discovered during heuristic evaluation. Usability issues were addressed and the latest iteration of the app presented no additional usability issues. | Heuristic evaluation |
| 4 | From PwMS, 68.9% indicated to intend to start using an MS app like icompanion, and 80.1% intended to start using it daily or weekly. Only one patient (2.2%) indicated to have a negative attitude towards using an app to monitor their disease course. | Survey |
| 5 | Compliance decreased over time in all cohorts. However, participation in elevateMS was generally consistent. The results demonstrate the feasibility and utility of a decentralized method to gather real-world data about participant's real-time life experience of MS through the app, as well as the importance of frequent, real-world assessments of MS disease manifestations outside of episodic clinical evaluations. | Task completion |

**Table 3.** *Cont.*

| Unique Identifying Number | Degree of Satisfaction | Usability Evaluation Method |
|---|---|---|
| 6 | The phone application showed overall initial feasibility, with only a small proportion of participants demonstrating noncompletion. So, the FatigueApp is a feasible way for collecting real-time symptoms of fatigue in patients with MS. | Task completion |
| 7 | N/A | N/A |
| 8 | Feasibility was assessed by questioning self-reported completion accuracy, smartphone or tablet skills, experiences and suggestions for improvement. Some subjects provided suggestions for improvement. | Questionnaire |
| 9 | Participants provided much positive feedback about MS Energize, and gave it reasonable ratings for usability. Participants rated MS Energize with a median total score of 75 on the SUS (mean score 72.3, range 65 to 90). | SUS; qualitative interview |
| 10 | The mean values for different sections of the questionnaire were between 6.1 and 9, indicating that the application was evaluated at a "good" level by the users. | QUIS questionnaire |
| 11 | N/A | N/A |
| 12 | The results prove that people evaluated WalkWithMe positively. The features were rated as useful, easy to use and understand, and attractive by most participants. All aspects regarding usability were evaluated positively by the participants in the self-made questionnaire. Impact overall received an average score of 3.4 out of 5. | uMARS questionnaire; self-made questionnaire; interview |
| 13 | N/A | N/A |
| 14 | N/A | N/A |
| 15 | N/A | N/A |
| 16 | In the device-usability questionnaire, all participants reported that it was easy or very easy to learn, use, navigate, read, and see the response choices of the eDiary. In the cognitive interviews, participants confirmed that the symptoms and impacts listed on the FSIQ-RMS were relevant to their experience with RMS. For that, the final FSIQ-RMS is a valid and reliable PRO instrument that has demonstrated content and measurement validity for fatigue-related symptom and impact items. | Questionnaire; Interviews |

Abbreviations: SUS: System Usability Scale; QUIS: Questionnaire for User Interaction Satisfaction; uMARS: user-versione of the Mobile Application Rating Scale; N/A: Not Available.

### 3.2.1. Questionnaires

Four studies used questionnaires for evaluating usability [20,23,26,29]. The usability of *MS Energize* was assessed through the SUS questionnaire followed by a qualitative interview. At completion of the test, participants rated MS Energize with an SUS median total score of 75/100, suggestive of positive feedback regarding MS Energize usability [26]. In a study by Mokhberdezfuli et al., the QUIS questionnaire was used to evaluate usability of the mobile application; the mean values for different sections of the questionnaire were between 6.1 and 9, indicating that the application was evaluated as "good" by users [29]. The usability of WalkWithMe was evaluated through the uMARS questionnaire, self-made questionnaires, and semi-structured interviews. Overall, the WalkWithMe application received an average score of 3.4 out of 5 (1 = strongly disagree, 5 = strongly agree), and all aspects regarding usability were evaluated positively by participants in the self-made questionnaire [23]. The usability of the new Fatigue Symptoms and Impact Questionnaire—Relapsing Multiple Sclerosis (FSIQ-RMS questionnaire) was evaluated through a device-usability questionnaire and cognitive interviews, confirming the conceptual equivalence of the FSIQ-RMS electronic-Diary (e-Diary) compared to the traditional version and its appropriateness for the use with RPwMS [20].

### 3.2.2. Task Completion

Four studies used task completion for usability evaluation [16–18,22]. Palotai et al. evaluated the usability of the smartphone application through the task completion method; specifically, the usability was assessed through the number of completed questions [18]. The study reported a high completion rate (84% of those who began the study) without significant correlation between adherence and variables, like age, sex, ethnicity, disease duration, EDSS, and FSS score [18]. Usability perception was good in Barrios et al: all patients completed the two-week protocol, and the adherence was good [17]. The usability of ElevateMS was evaluated through the assessment of compliance and task completion; while the compliance decreased over time, the participation was generally consistent [22]. The usability of FatigueApp was evaluated through the assessment of compliance and task completion. Most participants (87%) completed all the surveys; the dropout was highest in the last 4 days of the study time frame [16].

### 3.2.3. Heuristic Evaluation

Only one study used heuristic evaluation for usability assessment [31]. Heuristic evaluation includes the assessment of the application interface by a small group of users, testing usability through defined principles. Evaluators go through a website interface and perform different tasks to identify usability issues that need to be fixed for a smoother user experience [32]. More Stamina was evaluated for the design of the prototype and not for its use. A heuristic evaluation involving three Human–Computer Interaction (HCI) researchers was performed, and several usability problems were addressed; the latest iteration of the app presented no additional usability issues [31].

### 3.2.4. Think-Aloud and Questionnaire

The think-aloud protocol (TAP) involves the participant verbalizing their thoughts and literally saying everything that comes into their head while they are interacting with the technology [33]. TAP involves a one-to-one session between the user and the researcher; its purpose is to ask the participant to think aloud as a means to capture the problem-solving process [33]. Only one study used the TAP, followed by a questionnaire to evaluate usability, and the results suggest that the user experience of MS Energise was mostly positive [27].

### 3.2.5. Survey and Interview

icompanion was the only application whose usability was tested only through a survey [25], in that 68.9% of the sample showed interest in using icompanion [25].

MS Energize and WalkWithMe usability were assessed through interviews, showing positive results [23,26].

### 3.3. Feasibility, Reliability, and Fatigue-Related Results

Feasibility and reliability assessment was performed in 10 studies [16,17,19,21–25,28,30].

### 3.3.1. Feasibility Results

The icompanion has shown good feasibility in detecting the differences between MS subtypes and clinically relevant changes in cognition and fatigue [25].

ElevateMS showed the importance of frequent, specific assessments of MS symptoms outside clinical evaluations [22]. Tracking self-reported data allows one to identify the most common symptoms and triggers in PwMS, as well as significant associations ($p < 0.001$) between performance in active functional tests and disease severity [22]. Feasibility was also evaluated in FatigueApp [16]. The smartphone application showed overall initial feasibility to assess fatigue [16].

In the study of Barrios et al., the tapping task was used to discriminate between fatigued and non-fatigued participants, showing good feasibility [17]. Study results showed significant differences between the two classes of participants in terms of the mean tapping frequency holds in the wild for both questionnaires (FSS and FSMC). The authors confirmed that the tapping task is valid in unsupervised settings and suitable for monitoring fatigue remotely [17].

In Tapping and Balloon Popping Test, it was assumed that the decline in the top performance may represent an objective measure of fatigability [21]. In PwMS, an inverse correlation between fatigue and tapping speed was observed, indicating that PwMS with greater disability also experience greater motor fatigue [21].

In the study of Bove et al., the authors assessed the feasibility of data collection using the smartphone platform [19]. The deployment of several cognitive and motor tests through the smartphone platform should be a feasible way of overcoming the challenge of collecting ecological data that accurately describe the course of symptoms. The correlation between self-reported fatigue (assessed through the MFIS) and external factors was evaluated. Many PwMS showed fluctuations in perceived levels of fatigue over the year, but no significant evidence of a fixed effect of daylight hours or ambient temperature was found [19].

### 3.3.2. Reliability Results

The WalkWithMe app investigated the effects of the intervention in terms of energy used for an activity, showing an average increase of 2432 metabolic equivalent (MET) minutes per week [23]. During the use of the WalkWithMe app, an increase in energy was reported by patients, suggesting a positive contribution in terms of fatigue reduction [23].

The use of electronic visual analogue scales for recording fatigue, pain, anxiety, and quality of life in PwMS has proven to be reliable and useful, showing a high percentage of positive feedback by patients [24].

The reliability of the Neurokeys Keyboard APP was evaluated: keystroke dynamics allowed for highlighting patients and controls and was related to clinical disability measures [30]. However, no differences in timing-related keystroke features were observed between non-severely fatigued (MS-NF) and severely fatigued (MS-F) PwMS. FSS and Checklist Individual Strength Fatigue subscale (CIS-F) scores did not correlate with the composite keystroke features [30].

Finally, Barrios et al. (2020) assessed the reliability of a simple tapping task on a smartphone, an exertion technique to assess the user's motor fatigability compared to the standard handgrip dynamometer measure [28]. The study showed that participants' performance decreases during the tapping task and correlates with the decrease in grip strength measured with the handgrip dynamometer. The authors concluded that a 30 s tapping task is reliable to measure motor fatigability [28].

## 4. Discussion

The recent COVID-19 pandemic highlighted the barriers that people with chronic disease may encounter in access to medical treatment, limiting the possibility to receive adequate care [34]. In this regard, DHT, although distant from traditional rehabilitation paradigms, which are based on direct contact between patients and health professionals, represents a useful tool to improve patients' self-care and self-management, particularly in conditions in which access to health services is challenging [35,36]. For PwMS, especially in advanced stages, characterized by severe disability and exponential mobility challenges, the potential integration of conventional rehabilitation alongside home-based interventions would confer significant advantages for both the patient and the caregiver [37].

As far as we know, this scoping review represents the first attempt to describe the use of mHT for the management of fatigue in PwMS, one of the most reported symptom in MS with a significant impact on QoL [1]. The possibility to monitor fatigue in daily life can be extremely helpful for the patient in terms of empowering them and self-care. In the literature, the use of smartphone applications in PwMS was investigated, and 1042 applications were identified [13].

Our analysis was aimed at studying the usability, feasibility, and reliability of assessing and self-managing fatigue in PwMS using smartphone applications. Of note, almost all of the included studies were conducted in countries in the Western world (i.e., the USA, European countries or New Zealand), which probably reflects the different degree of investment and diffusion of new technologies and innovation for patient care, depending on the global region [38]. Indeed, numerous studies have reported that the development of DHT constitutes a distinct form of social inequity, since people who generally benefit most from these resources are also those who already have numerous advantages over other socio-economic and cultural groups in terms of health, e.g., higher incomes and, consequently, more availability for medical care, adequate lifestyle, and proper diet [14,39]. However, while some authors conclude that these technologies will lead to a further widening of the gap between high- and low-income countries, others suggest that DHTs could play a crucial role in addressing and reducing these disparities; for example, DHT could facilitate the collection of useful data to better understand disparities and provide tools to reduce the fragmentation of care for minorities and/or patients with limited English proficiency [40]. Consistently, a recent scoping review by Falkowski et al. documented that, despite numerous difficulties, even lower-middle-income countries are beginning to significantly develop these technologies to support their healthcare decision making [40]. Therefore, reasonably, DHTs will also lead to a major improvement in less-developed countries, and mHTs, due to their immediate availability and ease of use, could play a crucial role in this regard. However, in order to achieve this goal, it is also fundamental to support the validation of these apps in languages other than those currently commonly used, so as to also facilitate use by PwMS who are not familiar with, for example, the English language [41].

Although many studies did not provide crucial details from a methodological point of view, it is also important to note that DHTs have been evaluated in all MS subtypes (i.e., CIS, RRMS, PPMS, and SPMS) and with disease durations of more than 20 years, emphasising the possibility of their application, even in the most severe subtypes (in particular, progressive ones, although most studies evaluated RRMS, which is coherently the most frequent clinical phenotype) and when the expected disability is higher [42]. However, the maximum degree of disability, as assessed by means of the EDSS, was 5.31, i.e., a condition characterized by a modest impairment in daily autonomy; thus, patients with a severe disability were not assessed, probably also due to the frequent cognitive disturbances associated with the more advanced stages, which could compromise the cooperation and reliability of the provided answers [43].

Even if most of the described apps were developed for Android, some are also available on the App Store and others on both operating systems, a finding that suggests a growing interest in making these technological resources available to as many users as possible.

Importantly, each mobile app included offers different tools to evaluate fatigue symptoms. Specifically, all employed methods (i.e., questionnaires, interviews, survey, task completion, TAP, and heuristic evaluation, both alone or in combination) documented a high degree of usability, assessed both as feedback from participants and as the number of completed tasks, highlighting that these smartphone applications are significantly user-friendly for patients. Furthermore, through heuristic evaluation, it was also possible to identify and resolve users' concerns about the app, suggesting that an interface able to detect user feedback could be a very useful tool for improving patient compliance. Coherently, most of these mHTs have been developed according to a user-centred design, confirming the importance of that reported in the study of Gromisch et al., which suggests that including PwMS, as well as healthcare providers, during the design process can elucidate potential disease-related limitations and their solutions prior to the product being tested commercially [9]. Indeed, involving patients throughout the development of new apps, particularly in the early stages, can increase the likelihood that these tools are meeting the needs of end users and will be implemented in everyday life [9]. Since mHTs are an integral component of current telemedicine, our findings are in line with those obtained by Yeroushalmi's study, indicating that satisfaction with the level of care provided through DHT prompts a great potential for its widespread implementation [44]. Equally important would be the spread of these applications, whose use has proven to have high patient satisfaction.

Furthermore, the feasibility assessment also showed good results overall, as in most cases, the apps were able to discriminate between patients with and without fatigue in an objective and ecological way according to the patients' performance (e.g., by means of tapping tasks), also considering, in some cases, subjective evaluation and potentially relevant external factors. These results not only encourage a more widespread use of these remote assessment tools for PwMS but are also consistent with recent work in patients with other chronic neurological diseases, e.g., Parkinson's disease and stroke [45–47]. Considering the difficulty in the assessment of a subjective and insidious symptoms, such as fatigue, obtaining standardised and objective tools, also available at home, is crucial to improve the management not only of PwMS but also for numerous other pathologies, as it is extremely common in patients affected by chronic diseases [48,49].

Importantly, these apps also shown excellent results in terms of reliability in some studies, evaluated through simple tools, such as weekly walking minutes, visual analogue scales, and finger tapping exercises. Moreover, interestingly, some patients reported an improvement in their energy levels or a reduction in fatigue thanks to the use of mHT [23,24].

Taken together, these results confirm that most of the smartphone applications have been positively evaluated by patients, with a high degree of satisfaction in terms of ease of use and to actively participate in the management of their disease. Of note, the apps differed greatly from each other, both in terms of methods for assessing fatigue (e.g., self-report questionnaires, rating scales, *Patient Reported Outcome* (PRO) tools, and sleep diary), in the way the patient had to set and evaluate the achievement of daily targets (e.g., keeping a daily diary of the activities to be carried out, or to set a goal regarding the performance of physical activity, for example walking) and in the way the patient could improve symptomatic and therapeutic self-management (e.g., performing specific tasks to evaluate motor fatigability and motor and cognitive functions, and setting reminders for the therapy to be taken). While this makes it more difficult to compare the obtained results, it also highlights a wide range of possibilities in terms of mHT for PwMSs, suggesting that it might be useful to combine several apps to evaluate different aspects in a tailored way. In addition, in some apps, there was also the possibility to contact doctors, which certainly improved doctor–patient interactions and made it easier to deal with any doubts or concerns without the need for hospital access. Therefore, consistent with other work in the field, the benefits that PwMS perceived about using mHT included easy access to clinical information, communication with healthcare professionals and other patients, usefulness for monitoring MS, and major efficiency of MS administrative tasks [13]. Moreover, notably, ClinicalTrials.gov reports numerous recent clinical trials evaluating the use of mHT in

PwMS, and many are still recruiting patients, e.g., NCT05816122, NCT04953689, and NCT05865405. This indicates the current strong research interest in this field, considering the relevant implications it could bring to the complex management of these patients.

Despite the positive results that emerge from our research, some limitations should be considered when interpreting the results of this review. The choice of search strings, although essential for targeting results, could overlook relevant work in the literature. However, using four databases and the optimisation of the search parameters have, in our opinion, compensated for this methodological choice. Furthermore, this review was limited by the small number of studies that met the inclusion criteria, probably due to the still probably limited knowledge and research on this topic. It is also important to note that many studies did not report fundamental methodological information, such as sample description, e.g., sample size, the MS subtype, average disease duration, or EDSS, leading to a heterogeneous sample of studies. Moreover, some studies focused on the methodology of developing the app but did not carry out a clinical trial with an MS sample. Furthermore, not all studies performed an evaluation of usability and, in those that performed it, the assessment methodology was, most of the time, purely subjective. Indeed, most of the used tools were questionnaires, interviews, and surveys, which report only the subjective perception regarding the use and/or design of the smartphone application. In addition, in some studies, only the degree of patient satisfaction about mHT use was evaluated, but the application capability to guarantee adequate monitoring and management of fatigue was not evaluated. Some studies, instead, compared the use of simple tasks to be carried out on the smartphone (e.g., tapping task) with the standard method used in motor fatigability assessment (i.e., handgrip dynamometer) to search for an objective measure of fatigue assessment. However, it is not always clear whether the obtained results on motor fatigability are reliable with regard to the fatigue symptom characteristic of MS, which is characterized not only by motor symptoms but also cognitive ones. In this regard, further clarification would be needed. Finally, our work also highlights the wide variability in apps, which compounds the limitation of interpretability of the data, meaning it is not possible to generalize and use in clinical settings.

## 5. Conclusions

The world of DHT grows continuously, and more and more people are becoming adept at using these technological tools in daily life. Considering the benefit of this instrument in breaking down space-time barriers and reducing healthcare costs, and the importance for chronic PwMS to continuously manage their illness, the aim of this scoping review was to individualize and categorize smartphone-based applications for self-management of MS fatigue. These types of mHT are required to be designed based on the real needs of end users and the opinions of healthcare providers; therefore, users' requirements were investigated in most studies before designing the application. From the results of the studies included, it can be deduced that PwMS are satisfied with the apps and perceive a good usability of the tool. Improving the health status of PwMS through continuous monitoring and remote supervision by physicians, as well as a tool for treating motor and cognitive fatigue, is the greatest benefit we can achieve from these smartphone applications.

However, despite these tools constantly growing and transforming and that our results are strongly encouraging, the availability of information regarding this topic is still limited; thus, it is not possible to make clinical suggestions as the results are inconclusive. It follows that further clinical studies on the impact of this mHT on the QoL of PwMS, as well as the cost-effectiveness of smartphone apps, are recommended. Studies that report comprehensive sample information use standardised and reproducible evaluation tools (taking patients' and doctors suggestions into account during implementation) to assess usability, feasibility, and reliability, as well as specific management improvement measures (both as judged by the patient and the healthcare professionals), and offer the possibility of direct interaction with physicians. Indeed, the potential of DHTs and, specifically, mHTs is extremely high and should be properly exploited to improve the remote management and

the QoL of patients with fatigue, so as to improve the therapeutic management of MS in countries with reduced economic and health resources.

**Supplementary Materials:** The following supporting information can be downloaded at: https://www.mdpi.com/article/10.3390/sclerosis2010004/s1, Supplementary File S1: Search strategy.

**Author Contributions:** Conceptualization, G.M., I.C. and S.S.; methodology, A.A., A.B. and S.S.; investigation, A.B., G.M. and I.C.; data curation, A.A., A.B., G.M. and I.C.; writing—original draft preparation, A.A., A.B. and S.S.; writing—review and editing, A.A., A.B. and S.S.; supervision, A.A. and S.S. All authors have read and agreed to the published version of the manuscript.

**Funding:** This research received no external funding.

**Conflicts of Interest:** The authors declare no conflicts of interest.

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
