# Peer review of "Role of Smartphone Applications in the Assessment and Management of Fatigue in Patients with Multiple Sclerosis: A Scoping Review"

_2813-3064, doi:10.3390/sclerosis2010004_

Round 1
Reviewer 1 Report
Comments and Suggestions for Authors
The modern integration of Patient Reported Outcomes with Digital Health Technology, and the assessment of probably the most common MS symptom: fatigue, are reasonably address in your review despite the tremendous heterogeneity of the sources and the fact these tools are constantly transforming and being updated. (1) A comment is missing on the accessibility of these technological tools remaining limited to communities and countries able to afford them, in other words (and this is documented in your study), places with developed economies and adequate societal education and support to implement these systems. (2) The former issue also goes along with availability of the technology in different languages. You mention a few, including Spanish and cite a reference for such (number 22 in your list), however. that is a Belgian study, and this reviewer was unable to identify Spanish was being utilized in the surveys. Please clarify.
Author Response
The modern integration of Patient Reported Outcomes with Digital Health Technology, and the assessment of probably the most common MS symptom: fatigue, are reasonably address in your review despite the tremendous heterogeneity of the sources and the fact these tools are constantly transforming and being updated.
We thank the Reviewer for the positive feedback and for appreciating our work.
(1) A comment is missing on the accessibility of these technological tools remaining limited to communities and countries able to afford them, in other words (and this is documented in your study), places with developed economies and adequate societal education and support to implement these systems.
We thank the Reviewer for this important suggestion, we agree. Therefore, we modified the Introduction section as follows:
“Coherently, many smartphone applications for PwMS are commonly used and this field has undergone a steady growth in recent years, although, at least at the moment, the dissemination of these technologies is mainly limited to Western countries, i.e. those with the economic resources and educational level to support their implementation (Timmermans et al., 2020)”.
Moreover, in the Discussion section, we added the following paragraph (please see Discussion section):
“Indeed, numerous studies have reported that the development of DHT constitutes a distinct form of social inequity, since people who generally benefit most from these resources are also those who already have numerous advantages over other socio-economic and cultural groups in terms of health, e.g. higher incomes and, consequently, more availability for medical care, adequate lifestyle, proper diet (Timmermans et al., 2020; Shim, 2010). However, while some authors conclude that these technologies will lead to a further widening of the gap between high- and low-income countries, others suggest that DHTs could play a crucial role in addressing and reducing these disparities: for example, DHT could facilitate the collection of useful data to better understand disparities and provide tools to reduce fragmentation of care for minorities and/or patients with limited English proficiency (Lopez et al., 2011). Consistently, a recent scoping review by Falkowski et al. documented that, despite numerous difficulties, even lower middle-income countries are beginning to significantly develop these technologies to support their healthcare decision-making (Falkowski et al., 2023). Therefore, reasonably, DHTs will also lead to a major improvement in less developed countries and mHTs, due to their immediate availability and ease of use, could play a crucial role in this regard.”
Finally, in the Conclusions section, we added the following sentence (please see Conclusions):
“Indeed, the potential of DHTs and, specifically, mHTs is extremely high and should be properly exploited to improve the remote management and the QoL of patients with fatigue, also to improve the therapeutic management of MS in countries with reduced economic and health resources.”
(2) The former issue also goes along with availability of the technology in different languages. You mention a few, including Spanish and cite a reference for such (number 22 in your list), however. that is a Belgian study, and this reviewer was unable to identify Spanish was being utilized in the surveys. Please clarify.
We thank the Reviewer for emphasising this point. Indeed, the paper does not report information on the language (not only Spanish, but all the languages we reported) of the investigated mHT. In fact, precisely to collect this information, we consulted the icompanion MS site (please see https://icompanion.ms/) at the time of data extraction and reported the languages available. The Authors report only that "The survey was answered by 876 PwMS, predominantly located in the U.S. and Canada (91.4%)", and this implies a predominance (most likely) of assessments carried out in English and French, but most likely tests were also carried out in the other languages reported (e.g. between available European countries and US citizens from European regions). Ours was an attempt to highlight the available languages precisely in order to emphasise the importance of this aspect in fostering the spread of mHT. In fact, we agree with the Reviewer on the importance of this point and to emphasise it, we added the following to the Discussion (please see Discussion section):
“However, in order to achieve this goal, it is also fundamental to support the validation of these apps in languages other than those currently commonly used, so as to facilitate use also by PwMS who are not familiar with, for example, English language (Kruse et al., 2019).”
Reviewer 2 Report
Comments and Suggestions for Authors
This is interesting study describing the use of smartphone applications in assessment of fatigue in MS. I have just 2 comments - There is a lack of description of these techniques (even short) in introduction. Ale were they used in clinical trials?
Author Response
This is interesting study describing the use of smartphone applications in assessment of fatigue in MS.
We thank the Reviewer for the positive feedback and for appreciating our work.
I have just 2 comments - There is a lack of description of these techniques (even short) in introduction.
We thank the Reviewer for highlighting this point, we agree. Therefore, we have amended the Introduction as follows (please see Introduction):
“Although there are many types of mHTs in this field, they are generally commercially available smartphone applications that allow patients to actively participate in the management of their disease, not only by monitoring their daily routine through reports and diaries, but also by receiving treatment reminders and specific lifestyle advice from specialists (Giunti et al., 2017). In addition, they often provide access to simple exercises and games designed not only to assess one's performance over time, but also to improve it in a simple, fun and engaging way for the patient (Pless et al., 2023).”
And were they used in clinical trials?
We thank the Reviewer for this insightful suggestion, which allowed us to highlight even more the strong research interest in this topic. We performed a search on ClinicalTrials.gov and found several recent clinical trials on mHT in PwMS. Thus, we have added a few examples in the Discussion (please see Discussion):
“Moreover, notably, ClinicalTrials.gov reports numerous recent clinical trials evaluating the use of mHT in PwMS and many are still recruiting patients, e.g. NCT05816122, NCT04953689, NCT05865405. This indicates the current strong research interest in this field, considering the relevant implications it could bring to the complex management of these patients.”
Reviewer 3 Report
Comments and Suggestions for Authors
The work addresses the issue of using mobile devices to assess self-reported fatigue in patients with multiple sclerosis. It is a scoping review, which allowed us to conclude good levels of usability, validity and reliability, suggesting a promising role for this technology in this context
The introduction provides a good contextualization of MS, fatigue, and mobile technologies use
The methodology used is adequately described, the protocol was registered in the Open Science Framework, the objectives of the study are well explained (lines 72 to 80)
• "search, identify, and synthesize research on mobile applications for the assessment and the management of fatigue in PwMS.
• categorize the areas of assessment and/or intervention into psychological/cognitive and motor/functional.
• describe ability of mobile applications to reduce fatigue.
• describe users’ satisfaction and usability.
• provide clinicians a practical overview of the use of mHT in the assessment and management of fatigue in PwMS, highlighting the mobile applications that currently seem to be the most promising for this purpose."
Articles published in peer-reviewed journals, published conference proceedings, and pre-peer review web publications were potentially eligible. Searches of electronic bibliographic databases in Web of Science, PubMed, Science Direct and Embase from 2012 to 2022 inclusive were done.
The inclusion criteria are presented. The selection flow and intermediate results are presented and analysis of the work is presented. The work is summarized in a general table. Tables of studies using mobile phones are presented, describing the application and showing satisfaction analysis when available
A detailed analysis of the aspects of feasability and reliability in the various studies is also presented.
The discussion is detailed and uses current references
Reflection on the limitations of the study is appropriate
The conclusions are in line with the study
Author Response
We really thank the Reviewer for the very positive feedback and for appreciating our work.
Round 2
Reviewer 1 Report
Comments and Suggestions for Authors
None. The authors have replied adequately to my observations.